# HIV Vaccine Development at a Crossroads: New B and T Cell Approaches

**DOI:** 10.3390/vaccines12091043

**Published:** 2024-09-12

**Authors:** Ramesh Govindan, Kathryn E. Stephenson

**Affiliations:** 1Division of Infectious Diseases and Center for Virology and Vaccine Research, Beth Israel Deaconess Medical Center, Boston, MA 02215, USA; rgovind1@bidmc.harvard.edu; 2Harvard Medical School, Boston, MA 02115, USA; 3Ragon Institute of MGH, MIT and Harvard, Cambridge, MA 02139, USA

**Keywords:** HIV, vaccines, antibodies, T cells, B cells

## Abstract

Despite rigorous scientific efforts over the forty years since the onset of the global HIV pandemic, a safe and effective HIV-1 vaccine remains elusive. The challenges of HIV vaccine development have proven immense, in large part due to the tremendous sequence diversity of HIV and its ability to escape from antiviral adaptive immune responses. In recent years, several phase 3 efficacy trials have been conducted, testing a similar hypothesis, e.g., that non-neutralizing antibodies and classical cellular immune responses could prevent HIV-1 acquisition. These studies were not successful. As a result, the field has now pivoted to bold novel approaches, including sequential immunization strategies to drive the generation of broadly neutralizing antibodies and human CMV-vectored vaccines to elicit MHC-E-restricted CD8+ T cell responses. Many of these vaccine candidates are now in phase 1 trials, with early promising results.

## 1. Introduction

The last four decades have seen impressive strides in the prevention and treatment of human immunodeficiency virus (HIV) infection. The rate of new HIV infections has steadily declined each year, from a peak of 3.2 million people in 1995 to 1.3 million in 2022 [1]. In 2022, an estimated 86% of all people living with HIV knew their HIV status, and 29.8 million people were accessing antiretroviral therapy (ART). The demonstration of complete prevention of transmission with effective ART, represented by the concept of “undetectable equals untransmittable” (U = U), was a paradigm shift for both public health and individual outcomes [2]. This success was followed by the use of many of those same agents for pre-exposure prophylaxis (PrEP) [3]. The recent approval of long-acting cabotegravir and rilpivirine, as well as recent success of twice-yearly lenacapavir in preventing HIV-1 acquisition in cisgender women, brings hope for a brighter future via improved access and adherence [4,5,6].

Despite these advances, significant limitations remain. ART, while generally well-tolerated, can cause renal injury, mitochondrial toxicity, lipodystrophy, and neuropathy, requiring sustained clinical and laboratory monitoring [7]. For those who can tolerate ART, long-term toxicities, including weight gain and secondary metabolic syndrome remain of high concern. The emergence of ART resistance patterns and the complex interplay between viral mutational escape and ART interruptions has created an arms race between the virus and the research community. Perhaps the greatest shortcoming of current HIV therapies is that they do not constitute a viable cure. Long-lived populations of infected cells carry replication-competent proviruses (i.e., the “reservoir”) capable of viral resurgence and resistance within days to weeks of ART withdrawal [8]. Thus, even for those who can tolerate ART and achieve viral suppression, adherence is an indefinite, consistent commitment. In rare cases, stem cell transplantation has shown potential in curing HIV; the replacement of a patient’s immune cells with those from a donor who has natural resistance to HIV due to a mutation in the CCR5 co-receptor can eliminate the virus from the body [9,10,11,12]. These highly publicized cases (e.g., the “Berlin Patient” and the “London Patient”) represent the only current platform for curing HIV infection, but due to complexity, risk, and costs, it has not been translated to widespread use. Moreover, with the advent of ART and PrEP, the face of the struggle against HIV and acquired immunodeficiency syndrome (AIDS) has moved beyond the basics of virology, physiology, and pharmacology to also include the challenges of funding, infrastructure, and access. Approximately one-third of those who have HIV are undiagnosed, not in care, or not on effective therapy, and it is estimated that in 2022 thirty-nine million people were living with HIV and six hundred and thirty thousand people died from AIDS-related illnesses worldwide [1,13]. New PrEP options on the horizon, such as lenacapavir, will remain unaffordable for most people for the foreseeable future. 

In the face of this public health reality, the dream of an HIV vaccine remains alive and well, even in the setting of recent breakthroughs in PrEP. The potential benefits of an HIV vaccine are many. For example, the replacement of PrEP with a vaccine would obviate the need for adherence, monitoring, and testing for millions of individuals, reducing the strain on public health systems and drug supply chains. A recent study in adolescent girls and young women in sub-Saharan Africa demonstrated that an HIV vaccine remains the most preferred form of prevention, including over injectable PrEP, primarily because a vaccine has the potential to provide complete doses that offer protection for life [14]. Another study in the United States demonstrated a similar finding among men who have sex with men, e.g., that an HIV vaccine was preferable over injectable PrEP because it would not require indefinite repeated dosing and the associated costs [15]. These findings underscore how an HIV vaccine may have expanded uptake among individuals who had barriers to PrEP due to access and cost. Moreover, an HIV vaccine does not have the risk of eliciting antiretroviral resistance among the populations most at risk for HIV and therefore in need of ART in the future.

Inspired by the potential of an HIV vaccine to dramatically reduce disease burden, multiple international funding agencies and partners continue to commit substantial resources towards developing an effective and affordable HIV vaccine, including the National Institute of Allergy and Infectious Diseases (NIAID) and its HIV Vaccine Trials Network (HVTN), the Bill and Melinda Gates Collaboration for AIDS Vaccine Discovery, and the International AIDS Vaccine Initiative, as well as many others. Unfortunately, in recent years, there have been a series of disappointments in the quest for a safe and effective HIV vaccine, but those same setbacks have yielded insights that fuel further innovation. Here, we summarize major recent efforts towards HIV vaccine development and discuss novel platforms for further investigation.

## 2. Foundational Studies in HIV Vaccine Research

Vaccination, in its simplest terms, is the delivery of a foreign antigen into the body to stimulate an immune response that exhibits both sensitivity to and memory of that antigen. The adaptive immune response has two composite halves: cell-mediated immunity, facilitated by the body’s T lymphocytes, and humoral immunity, manifested in circulating soluble antibodies produced by B lymphocytes. These antibodies are polyclonal, meaning that they exhibit significant sequence, structural, and functional diversity, and, in the context of HIV-1, can be broadly divided into non-neutralizing, neutralizing, and broadly neutralizing antibodies. Neutralizing antibodies (nAbs) are antibodies that bind and directly inhibit the viral envelope (Env), the glycoprotein on the HIV surface that facilitates viral–host cell membrane fusion to initiate cellular infection [16]. The mechanisms by which nAbs act to inhibit Env are varied and are discussed here later. In contrast, non-neutralizing antibodies (nnAbs) also bind Env but do not inhibit its function; instead, they can affect Env activity through the fragment antigen-binding (Fab) region activity to induce viral aggregation or capture or via fragment crystallizable (Fc) effector functions, such as complement activation and antibody-dependent cellular cytotoxicity (ADCC). Finally, broadly neutralizing antibodies (bnAbs) are those antibodies that directly inhibit Env function across multiple HIV clades. bnAbs have recently come to light as a new holy grail for HIV vaccine design. The shift in emphasis from nnAb toward bnAb generation has come about through the efforts of multiple major trials over the past fifteen years (summarized in Table 1).

The first trial to demonstrate an HIV vaccine with a statistically significant protective effect was the 2009 Thai Prime-Boost/RV144 trial [ClinicalTrials.gov number NCT00223080] [17]. This trial studied a regimen of four priming injections of a recombinant canarypox virus vector vaccine (ALVAC-HIV) with two booster injections of a recombinant protein-based vaccine consisting of gp120, a subunit of Env. Notably, the gp120 protein component was designed to elicit nAbs to HIV, and its use as a single-agent vaccine had been explored previously, showing no protection in a cohort of Thai individuals engaged in injection drug use [18,19]. In RV144, roughly sixteen thousand healthy young men and women in eastern Thailand received a four-dose vaccine series over twenty-four weeks and underwent surveillance for 3.5 years. A major advantage of this study was that the HIV epidemic in Thailand was relatively homogenous during this time frame (HIV strain CRF01_AE), and the vaccine was serologically well-matched to the population. And indeed, in the final analysis at 3.5 years, a modest protective effect of the vaccine was revealed, at 31.2%. Follow-up studies using serum from human subjects of RV144 showed little to no neutralizing antibody activity, suggesting that protection was mediated by non-neutralizing Fab- and Fc-mediated functions [20,21,22,23]. Case-control correlate analyses suggested that this protective effect was mediated by antibodies specific to V1V2 variable loop epitopes on Env, which are located at the apex of the glycoprotein and exhibit significant sequence diversity [24,25]. These findings sparked a decade of studies in both humans and non-human primates, supporting the role of V1V2-specific antibodies in an anti-HIV immune response [26,27,28,29,30,31,32,33,34], as reviewed by Zolla-Pazner 2019 [35]. Interestingly, neutralizing antibody generation was not a correlate of protection in RV144, and these subsequent studies consistently demonstrated a lack of neutralizing antibody generation. Altogether, the results of RV144 fostered a new paradigm in which the generation of nnAbs capable of Fab- and Fc-mediated functions was hoped to protect against HIV.

The success of RV144 led to the establishment of the Pox Protein Public–Private Partnership (P5), which, in 2016, began HVTN 702, a prime-boost regimen study across fourteen sites in South Africa [NCT02968849] [36]. Instead of the subtype E virus used in RV144, study designers utilized a recombinant canarypox vector and protein boost generated from HIV-1 subtype C sub-Saharan strains that are more relevant to the circulating strains in South Africa. In contrast to RV144, in which subjects were vaccinated over a six-month interval, the 5404 participants in HVTN 702 were vaccinated over eighteen months, with the primary outcome of occurrence of HIV-1 infection at 24 months. Unfortunately, at 24 months no significant effect of vaccination was evident, with 138 infections in the vaccine group and 133 in the placebo group, and the trial was concluded early. Immunogenicity analyses of the HVTN 702 regimen, as compared to the RV144 regimen, studied in HVTN 100 [NCT02404311] and HVTN 097 [NCT02109354] trials, showed higher antibody binding and T cell responses to vaccine-matched peptide pools with the HVTN 702 regimen, but lower antibody responses to the V1V2 region, with overall similar levels of ADCC activity [37,38]. Comparing the results of RV144 and HVTN 702 is difficult; differences in vaccine components, immunization regimens, HIV-1 incidence, viral genetic diversity, and even host HLA genotype all have been suggested to play confounding roles (reviewed by Zolla-Pazner Lancet 2021 [39]).

Supported by the example of a viral vector-based vaccine in RV144, a public–private partnership between Janssen, the Gates Foundation, NIAID, and academic partners sponsored the Imbokodo HVTN 705 [NCT03060629] and Mosaico HVTN 706 [NCT03964415] trials. With the goal of eliciting greater immune breadth to circulating HIV-1 strains, researchers developed bioinformatically optimized bivalent global mosaic Env and gag-pol immunogens and expressed them on an adenovirus serotype 26 (Ad26) vector [40,41]. In a simian–human immunodeficiency virus (SHIV) challenge model in rhesus macaques, this vaccine regimen had offered 67% protection against infection, and in phase 2 trials it showed increased breadth of response to multiple V1V2 variants [41,42]. Unfortunately, both Imbokodo, which studied the efficacy of the vaccine among South African women, and Mosaico, which studied the same among men who have sex with men (MSM) in the Americas, were stopped early due to futility at interim analyses [43]. While immune correlate analysis showed a trend of IgG3 V1V2 breadth as an inverse correlate of risk, the robust immune responses seen in preclinical studies were not replicated [42].

## 3. Broadly Neutralizing Antibodies and the AMP Trials

The antibody-mediated neutralization of HIV relies on the ability of an antibody to directly bind to and inhibit the function of the HIV Env ‘spike’, the heavily glycosylated fusion protein on the surface of the HIV virion. Env is a trimer of heterodimers—each monomer consists of the noncovalently associated proteins gp120 and gp41. The gp120 ‘head’ domains primarily engage the host cell receptor CD4, triggering a conformational change in Env to expose an adjacent region of gp120 to bind the secondary host cell receptor (either CCR5 or CXCR4). This triggers a chain of dramatic, dynamic rearrangements that allow the gp41 ‘stalk’ domain to coordinate the fusion of the viral envelope and the host cell membrane. Subsequently, viral contents, including the RNA genome, are delivered into the cell. Env is a member of the Class I viral fusion glycoproteins, along with influenza hemagglutinin, Ebola glycoprotein, and SARS-CoV-2 spike protein, which exhibit the same global structural characteristics and exhibit a great deal of functional conservation.

Despite this functional conservation, Env displays an enormous amount of sequence diversity across the nine major (‘M’) HIV-1 clades. Based on the amino acid sequence alone, each Env clade differs from the others by about 30%, and sequence diversity within a clade is as high as 20% [44]. HIV-1 is able to rapidly generate mutant forms due in part to its large population size, its rapid replication rate, and the error-prone nature of the viral reverse transcriptase [45]. This gives it an ability to evade both antiviral therapy and immune control. Furthermore, heavy glycosylation and steric shielding of conserved functional domains mean that most naturally generated antibodies against Env are against regions of gp120 or gp41 that are not involved in fusion function [46]. The discovery of sera from patients that were able to neutralize primary isolates of HIV-1 in vitro led to the first mapping of vulnerable epitopes on Env; the identification of the first bnAb against HIV-1, b12, was an important milestone in HIV vaccinology [47]. The b12 antibody binds to the CD4 binding site (CD4bs) on gp120, preventing receptor attachment, and thus neutralizing Env function. Subsequent years saw the discovery of many other bnAbs, forming the “first generation” of HIV bnAbs. These include more that bind the CD4bs, others, such as 2F5 and 4E10, which bind the membrane-proximal external region (MPER) on gp41, and 2G12, which binds oligomannose residues on gp120 [48].

Early studies dating from the late 1990s [49,50,51] established that “firstgeneration” bnAbs could prevent the acquisition of simian–human immunodeficiency virus (SHIV) in non-human primates, beginning with pivotal studies showing that the passive infusion of the antibodies 2F5 and 2G12 could block SHIV infection following both intravenous and vaginal challenges [50,52]. As more potent bNAbs have been developed, dozens of non-human primate studies have been conducted that confirm that passive bNAb infusion can block SHIV acquisition (reviewed in Pegu et al. [53] and Julg and Barouch [54]). One of the first studies with newer generation bNAbs was conducted by Moldt et al. in 2012, showing that PGT121, which is directed against the V3-glycan, could achieve sterilizing immunity at lower doses than had been demonstrated with first-generation bNAbs [55]. Further studies on PGT121 showed that passive bnAb infusion promotes innate immune system activation in response to viral challenge and promotes viral clearance from mucosal tissue [56]. A similar study utilizing 10-1074, another V3-glycan-directed bnAb, demonstrated durable protection in a non-human primate rectal challenge model [57]. Nevertheless, follow-up studies have shown that combinations of antibodies are necessary for protection against a mixture of diverse SHIVs [58,59].

The 2021 Antibody Mediated Protection (AMP) trials HVTN 703 [NCT02568215] and HVTN 704 [NCT02716675] were the first proof-of-concept studies in humans of the protective efficacy of a single bnAb (Table 1). In these parallel trials, at-risk individuals without HIV received passive immunization with VRC01, a bnAb against the CD4bs on HIV-1 Env, every 8 weeks for 20 months, with a total of 4623 individuals participating across four continents. Neither trial was successful, in that overall VRC01-mediated protection was insignificant. Further analysis, however, revealed a successful proof-of-concept—when isolates of HIV-1 acquired during the trial were stratified by in vitro sensitivity to VRC01, it was discovered that passive immunization with VRC01 did in fact reduce the transmission of viruses sensitive to VRC01 (defined as those with an 80% inhibitory concentration of less than 1 µg/mL). For these viruses, the protective efficacy of this passive immunization was 75% [60]. The importance of this should not be understated—the demonstration of passive humoral immunity as providing transmission protection, even without the concomitant induction of anti-HIV-1 CD8 T cell responses, was a landmark moment that sparked a portfolio of follow-up passive immunization studies. Furthermore, the AMP trials established serum antibody neutralization titer as a correlate of protection against HIV-1 acquisition [61].

Many additional studies have taken the form of analytical treatment interruption (ATI) trials, in which individuals living with HIV are taken off of ART to assess the therapeutic effect of neutralizing antibodies. Among these antibodies was VRC07-523LS, a variant of VRC01 with improved potency, breadth, expression, and biophysical properties, further modified to prolong serum persistence and increase mucosal tissue localization. After showing safety in two phase 1 clinical trials, VRC 605 [NCT03015181] and VRC 607 [NCT02840474], it is being investigated in phase 2 ATI trials now [62]. 3BNC117, another bnAb targeting the CD4bs, was studied extensively and, in the MCA-0835 trial [NCT02018510], maintained viral suppression for up to 28 days in ART-naive, aviremic individuals living with HIV [63], enhancing host humoral responses to HIV-1 [64]. The MCA-0885 phase 1 dose-escalation trial [NCT02511990] studied the bnAb 10-1074, which targets the V3 glycan site on Env. High potency was observed, but unfortunately, but unfortunately there was also the induction of a high frequency of fully-resistant the induction of high frequency of fully resistant escape variants [65]. PGT121, another bnAb against the V3 glycan site studied in IAVI T001 [NCT02960581], demonstrated viral suppression for over 168 days in two subjects [66]. Unfortunately, as with the first-generation bnAbs, viral suppression from bnAb infusion was short-lived, with viral rebound driven by the emergence of resistance mutations or simply by decay in bnAb titer. Building on logic applied to ART, further studies using combinations of two or even three bnAbs to provide increased antiviral breadth, as well as combinations of bnAbs with immunostimulatory agents, have been undertaken to more thoroughly investigate the protective potential of interval bnAb infusions to protect people at risk of contracting HIV [67,68,69,70]. In the therapeutic context, the simultaneous infusion of multiple antibodies restricted viral escape, leading to longer-lasting viral suppression. Furthermore, the use of different classes of antibodies, such as CD4bs-targeting 3BNC117 and V3 glycan site-avid 10-1074, worked more effectively together in ATI viral suppression compared to either antibody alone [68,71].

Establishing the target concentrations for bnAbs to achieve efficacy using pharmacokinetic and interaction models with in vitro potency data has become a focus of the field [72]. Furthermore, the potential for bi-specific and tri-specific antibodies, i.e., a single protein merging multiple bnAbs, is also under investigation [73,74]. Thus, passive immunization strategies remain a relevant avenue of inquiry to provide protection to at-risk groups, with significant work being performed by the International AIDS Vaccine Initiative (IAVI) and NIAID, among others, to identify and study bnAbs from populations across the globe.

## 4. B Cell Approaches to HIV Vaccine Development

The demonstration of bnAb-mediated protection in the AMP trials established renewed urgency and interest in generating vaccines capable of eliciting bnAbs, particularly ones that target the CD4bs, V3 glycan supersite, the V2 apex of the Env trimer, and the membrane-proximal external region (MPER). Further neutralizing antibodies against the CD4bs, now termed ‘VRC01-class antibodies’, have been isolated from at least ten donors, and their structural characterization has identified specific conserved residues involved in binding the CD4bs [75,76,77]. Simultaneously, there has been a recognition that bnAb generation during natural infection is rare, likely requiring months to years of chronic uncontrolled infection, the stimulation of rare bnAb germline precursor B cells, and the engagement of virions that express Env antigens with the structural and conformational features required to optimally engage those precursor B cells [78]. In broad strokes, B cells begin differentiation via a complex process known as VDJ recombination within the B cell receptor (BCR) gene, which generates several naive B cells, each with the ability to recognize a specific, random antigen. Stimulation of a specific naive BCR by an antigen triggers affinity maturation, a process in which the stimulated B cell travels to a lymphatic germinal center, and its BCR precursor matures over 1–2 years via largely random somatic hypermutation and iterative antigen binding. In each iteration, CD4+ T helper cells assist in the survival of BCR mutants with higher antigen affinity and avidity. The end-product of this cycle is, ideally, a broadly neutralizing, high-avidity BCR that becomes the secreted bnAb. Studies of identified HIV-1 bnAbs indicate high levels of mutation (upwards of 30%) from their naive BCR precursors, suggesting a high requirement for somatic hypermutation and antigen-driven selection, particularly in those that bind the CD4bs [79,80,81]. Xiao et al. hypothesized that this high degree of somatic hypermutations may make it difficult for Env to bind germline antibodies, and that novel vaccine immunogens may need to be designed that can bind intermediates in the pathways to maturation [81]. Antibodies isolated from patients identified as “elite neutralizers”, i.e., those patients with broad and potent HIV-1 neutralizing antibody responses, have provided starting points for structure-based immunogen design [75,82].

In 2023, Martin et al. put forth a conceptual model outlining three overlapping strategies to guide B-cell maturation towards bnAb production: 2‘mutation guided immunogen design’, ‘structure-based immunogen design’ and a ‘germline/lineage agnostic immunofocusing approach’ [83] (Table 2).

The mutation-guided immunogen design approach utilizes sequential immunization with Env ‘booster’ antigens to guide the BCR affinity maturation trajectory [79]. The design of these Env booster antigens is informed by studies of virus and B cell co-evolution over the course of natural infection, during which antibodies against HIV-1 are isolated from the donor and temporally matched to the same donor’s BCR sequences, beginning with the BCR unmutated common ancestor (UCA) and the paired transmitter/founder (T/F) virus. In this way, researchers have been able to identify structural changes in the Env antigens that elicit the production of increasingly potent nAbs over the affinity maturation trajectory [80]. Intriguingly, these studies have revealed the existence of other, less-neutralizing antibodies that develop early in infection and exert selective pressure on the viral reservoir, generating escape mutations that in turn become epitopes for bnAb production [80,84]. The importance of diversity in the viral reservoir in B cell maturation, i.e., the necessity of a myriad viral antigens to stimulate bnAb BCR precursors, is not well understood. The observation that viral sequence diversification immediately precedes the development of antibody neutralization breadth in natural infection suggests either that there is a 1:1 association between a specific Env immunogen and a bnAb BCR precursor, or that the sampling of multiple Env variants during affinity maturation allows for BCR refinement towards broadly neutralizing activity [78,79].

Mutation-guided immunogen design is the basis of a series of clinical trials in which repeated immunization with a soluble Env construct is hoped to elicit a bnAb (Table 2). The production of soluble Env constructs has proven scientifically challenging; the metastable conformation in which the trimeric protein is held is one that is prone to collapse, and the presence of significant hydrophobic transmembrane domain presents a challenge for synthesis and purification [85]. One of the early solutions to these challenges was the development of the SOSIP trimer, in which a disulfide bond (SOS) was created between gp120 and gp41 to stabilize the trimer, and an isoleucine was changed to a proline at residue 559 to increase the affinity between the gp41 subunits in the trimer. One such construct, termed BG505 SOSIP.644, approximates the virus sequence BG505, but with modifications to create epitopes for several bnAbs [85]. This construct was demonstrated in vitro to have high avidity towards many known bnAbs, and has been taken forward into a series of clinical trials. The phase 1 HVTN 137 [NCT04177355], which aims to study BG505 SOSIP.644 in adults without HIV, is underway; preliminary data, published as a preprint, report that the trimer (adjuvanted with 3M-052-AF/alum) elicited robust, trimer-specific antibody, B-cell and CD4+ T cell responses, with five vaccinees developing serum autologous tier-2 nAbs [86]. The isolation of the CD4bs-binding bnAb CH103 from a patient in Africa led to the HVTN 115 trial [NCT03220724]. This patient, CH505, was followed through their infection from the initial acute phase until bnAb development, which, in this case, took roughly fourteen weeks. The isolation of the UCA of the CH103 bnAb lineage, as well as the CH505 T/F Env, the initial driving antigen for the CH103 UCA, provided a starting point for affinity maturation [80]. In HVTN 115, the CH505 T/F Env trimer was used as a priming immunogen in healthy adults, but unfortunately it was not shown to expand CH103 precursors. A follow-up study, HVTN 300 [NCT04915768], is underway using a CH505 T/F trimer construct with higher affinity for the CH103 UCA [87]. Similarly, DH270, a bnAb which binds the V3 glycan, was isolated from a patient, CH848, living with HIV in Malawi, and the identification of its UCA led to the HVTN 307 trial [NCT05903339] [88]. The study of the co-evolution of the DH270 UCA with the Env reservoir, including cryo-electron microscopy to visualize the interactions between the DH270 clonal tree and the co-evolving viral Env, has led to major insights in the affinity maturation process [89].

Parallel work on immunogen design has informed the structure-based immunogen design approach for bnAb production (Table 2). While the mutation-guided approach selects a priming immunogen identified to have affinity for the UCA of a known bnAb, the germline-targeting approach seeks to identify a priming immunogen that binds diverse precursors within a bnAb class that spans many lineages [90]. Here, the VRC01-class antibodies guided initial efforts, with the development of inferred germline (iGL) versions of these bnAbs through computational homology-guided reversion into their chromosomally templated sequences. These iGL bnAbs are hoped to approximate the naive bnAb-destined BCR precursors prior to affinity maturation. Interestingly, these iGL precursor bnAbs display neither reactivity to recombinant Env nor neutralization activity against HIV-1 isolates [77]. The reverse engineering of Env constructs designed to engage the iGL bnAbs has created a new class of ‘germline-targeting’ immunogens hoped to stimulate naive precursor B cells [91,92].

One such immunogen, eOD-GT6, was published in 2013 and has the ability to activate both germline and mature VRC01-class B cells [91]. The eOD, or “engineered outer domain” is in fact a modified CD4bs with both affinity for predicted iGL bnAbs and high affinity for the VRC01-class bnAbs to help guide affinity maturation towards the mature bnAbs [91]. This was followed shortly by eOD-GT8, which displayed picomolar affinity for the iGL bnAb of VRC01 [93]. These immunogens were engineered to form self-assembling nanoparticles displaying dozens of copies of the antigen along the outer surface. An important milestone in the field of germline-targeting immunogen design, the IAVI G001 phase I human trial [NCT03547245], showed that eOD-GT8, adjuvanted with AS01B, induced VRC01-class IgG B cells with substantial frequencies in blood and lymph nodes [90]. Enrolling forty-eight individuals, the participants were administered vaccine at 0 and 8 weeks, and, via B cell sorting and receptor sequencing, the researchers were able to detect the induction of CD4bs-specific IgG memory B cells after the first immunization, which increased further after the second dose.

The germline/lineage agnostic, immunofocused strategy of vaccine design is a simpler concept involving immunization with an epitope of interest that has been separated from the rest of the envelope (Table 2). On the Env glycoprotein, the MPER in particular, is an intense locus of epitope-focused vaccine design. Distinct from other classes of anti-Env bnAbs, MPER-targeting bnAb epitopes include the virion lipid membrane, as well as amino acid epitopes on the gp41 core [94,95]. For 2F5, one of the first-generation bnAbs isolated against the MPER, that epitope is a specific stretch of amino acids (ELDKWA) in the proximal region of gp41, and 4E10 binds a more distal NWFDIT epitope [96]. The finding that several MPER-targeting bnAbs displayed autoreactivity to self-lipids and the tryptophan enzyme KYNU spurred interest in the design of more specific immunogens capable of eliciting these bnAbs through the tolerance checkpoints of B cell maturation [96,97,98]. Based on studies of 2F5, gp41 immunogens have been developed that specifically elicit MPER-targeting bnAbs [96]. The fruition of this work came in 2024, with the results of the HVTN 133 clinical trial, which demonstrated induction of polyclonal HIV-1 B cell lineages by a MPER peptide-liposome [NCT03934541] [99]. Designed to bind an unmutated ancestor antibody of 2F5, a MPER peptide-liposome immunogen was tested in 20 participants at low risk for HIV acquisition. Two immunizations elicited a 95% serum binding antibody response rate to the immunogen, with a 100% peripheral blood CD4+ T cell response rate. The B cell repertoire analysis showed the presence of both neutralizing antibody B cell clones and those of their precursors. This study provided templates upon which further immunogen design could be based, with work underway to design mRNA-based MPER immunogens [99].

The recognition that these germline-targeting immunogens also generated non-neutralizing antibody responses led to the development of anti-idiotypic monoclonal antibodies (ai-mAbs) that bind and stimulate iGL precursors [100,101]. The benefit of these ai-mAbs is that they are not based on Env constructs, but are instead designed against the iGL Fabs, and they are therefore predicted to be less likely to generate off-target anti-Env antibody responses. These immunogens have been pursued even further to develop bispecific immunogens derived from multiple ai-mAbs [77]. Preclinical studies have shown that vaccination can induce broadly neutralizing antibody precursors to CD4bs, the V3 glycan supersite, and gp41 [102,103,104,105,106].

Part of the paradigm of the germline targeting approach is that once the appropriate precursor B cells are primed, affinity maturation should be further “shepherded” or guided towards bNAb development with iterative boosting immunogens, with a final immunogen to “polish” or optimize the immune response for neutralization breadth, potency, magnitude, and/or durability [106]. From a practical perspective, it is difficult to design, manufacture, and test new immunogens on the rapid timeframe necessary to test these iterative approaches. One potential solution is the use of mRNA-based vaccines similar to those used effectively for the prevention of COVID-19, with two clinical trials leading the way: HVTN 302 and IAVI G002. In HVTN 302, a BG505 MD39 native-like trimeric Env liposomal mRNA is being tested in 108 individuals in a phase 1 clinical trial [NCT05217641]. In IAVI G002, a phase 1 trial of eOD-GT8 60mer mRNA is being tested [NCT05001373]. These studies are ongoing, but one setback has been identified: an unusually high rate of pruritic skin rashes [107]. Further studies to elucidate the cause and frequency of these skin findings are underway, and other mRNA HIV vaccine constructs continue to be studied.

An additional challenge for iterative vaccine development is identifying the boosting immunogens that will be able to shepherd or guide B cell affinity maturation. One novel approach is to study the immune responses of people living with HIV who are undergoing an analytic treatment interruption. For example, in the HVTN 807 trial [NCT06006546], the participants are immunized with 426.Mod.Core-C4b adjuvanted with 3M-052-AF/alum. One aim of the study is to identify viral Envs that emerge during treatment interruption that may guide the maturation of the germline VRC01 responses elicited by the germline-targeting Env immunogen. Treatment interruption studies are being posited as a potential tool in the HIV vaccine field in general, particularly as it becomes necessary to have very large sample sizes for efficacy trials, given the frequent coincident use of highly effective PrEP among study participants.

## 5. T Cell Approaches to HIV Vaccine Development

The challenge of engineering immunogens to elicit an effective B cell response makes it necessary to also pursue vaccines that aim to utilize cellular immunity to suppress and potentially clear viral infection once acquired (Table 2). Prior HIV vaccine strategies relied on non-persistent vectors that produced antigens for a limited period, stimulating central memory TCM cells that are not usually pre-positioned at mucosal sites to mount effector functions [108]. For example, the phase 3 HVTN 502/Step and 503 trials [NCT00095576] tested a T cell vaccine based on an adenovirus 5 vector expressing subtype B HIV-1 Gag, Pol, and Nef proteins but did not demonstrate efficacy in high-risk participants in the Americas, Australia, and South Africa [109,110]. Moreover, follow-up analyses suggested an increase in infections in Ad5-seropositive or uncircumcised men, raising a concern that the Ad5 vector may influence susceptibility to HIV acquisition [111]. As with all HIV vaccine studies, the challenge for T cell vaccines is generating cellular immune responses that can protect against the wide range of global circulating HIV-1 strains [112]. Bioinformatic tools have been leveraged to better design T cell immunogens, including mosaic immunogens that can elicit a breadth and depth of cellular immune responses, as well as immunogens that contain highly conserved epitope sequences that might be shared by a range of viruses, such as the HIVACAT T cell Immunogen, HIVConsVx, and MVA.HIVconsv [113,114,115,116,117,118]. Many of the T cell approaches, particularly those for conserved element immunogens, have been tested primarily in the therapeutic context (reviewed in Brander and Hartigan-O’Connor in Curr Opin HIV AIDS) [119]. However, the mosaic approach was tested in multiple phase 2 and 3 preventive vaccine clinical trials, including TRAVERSE [NCT02788045], APPROACH [NCT02315703], and ASCENT [NCT02935686], which used an adenovirus 26 vector to express mosaic Env, Pol, and Gag proteins in order to elicit both B and T cell responses [41,120,121]. Unfortunately, the mosaic approach did not achieve efficacy against HIV-1 protection in the phase 3 Imbokodo and Mosaico trials discussed previously, despite eliciting diverse T cell responses [43].

Vaccination with viral vectors that provide a controlled, persistent level of antigen expression has been found to stimulate and maintain functionally differentiated effector memory T_EM_ cells in tissues beyond lymphoid sites [108,122,123]. These data suggest that a live, attenuated viral vector that could establish low-level, persistent expression of HIV antigens may confer unique immunologic benefits, leading to HIV protection or control. One such viral vector under investigation is based on an attenuated strain of human cytomegalovirus (HCMV), similar to those used in clinical trials of HCMV vaccines [124,125,126]. To explore the potential of HCMV as an HIV vaccine platform, investigators designed a rhesus cytomegalovirus (RhCMV) vector based on an available strain (68-1) and modified it to express the simian immunodeficiency virus (SIV) antigens [127,128]. As determined later, the 68-1 strain contains deletions of genes encoding proteins of the pentameric complex, designated UL128-UL130, which were retained in prior candidate HCMV vaccine strains. These deletions resulted in the vaccine eliciting non-canonical HLA-E-restricted CD8+ T cells. In multiple experiments over the last decade, the 68-1 RhCMV/SIV vaccine was shown to protect 50 to 60% of vaccinated rhesus macaques, exhibiting early and complete replication arrest and subsequent clearance of a highly pathogenic SIV challenge virus [127,129,130]. This protection was observed across different routes of infection (rectal, vaginal, parenteral), durable up to 3 years following vaccination, and was not abrogated by further deletions to enhance attenuation.

The protection elicited by the RhCMV/SIV vaccine does not follow any previously described pattern observed in preclinical models. RhCMV-SIV-vaccinated NHPs were repeatedly challenged with the highly pathogenic SIVmac239 strain until all animals demonstrated evidence of infection, defined as either viremia and/or development of a de novo Vif-specific T cell response, an antigen contained in the challenge strain but not in the vaccine. Remarkably, ~20% of the infected animals never had documented viremia, and the remaining protected animals had relatively transient and low-level viremia. When the protected animals were followed longitudinally, in addition to clearance of viremia, they had no latent SIV infection detected at necropsy, as assessed by ultrasensitive polymerase chain reaction (PCR) of tissues, viral co-culture, and the inability to transmit virus through adoptive transfer experiments to naive animals [129]. This “control and clear” mechanism of protection contrasts with elite controllers or ART-suppressed animals which, despite being aviremic, continue to harbor replication-competent virus. The HCMV-HIV vaccine candidate VIR-1111 was studied in 27 healthy individuals, with data analysis pending [NCT04725877]. The vaccine candidate VIR-1388 is being investigated in 95 individuals in the HVTN 142 study, with enrollment ongoing [NCT05854381]. One additional innovation of the VIR-1388 vaccine candidate is the addition of an mfuse1 insert, which is intended to generate more diverse HIV-1-specific cellular immune responses [131].

Other novel approaches are under study to elicit increased diversity of T cell responses with HIV vaccination. For example, in HVTN 085 [NCT01479296], Miner et al. studied the polytopic and fractional administration of a recombinant Ad5 vector expressing Gag, Pol, and Env sequences from three sub-types: ‘polytopic’ administration meant that injections with the different subtype-specific inserts were given in four separate anatomical locations, while ‘polytopic fractional’ meant that all the subtypes were mixed together, but they were given at one-fourth the dose at each of the four anatomical sites [132]. The authors found that polytopic fractional delivery resulted in a higher rate and magnitude of CD8+ T cell responses, as well as the greatest breadth of epitope-specific T cell responses. Ultimately, these T cell approaches may need to be combined with the B cell approaches discussed above to be most effective [133]; for example, one strategy is to have a layered approach, wherein breakthrough HIV infections that make it past the first line of bNAb defense are subsequently suppressed and cleared by back-up cellular immune responses [131].

## 6. HIV Vaccine Challenges beyond Immunogen Design

The scientific barriers to developing an effective HIV vaccine are described in detail above and underpin the complex immunologic B and T cell approaches that are being pursued. However, other challenges remain for an HIV vaccine, which will also need to be addressed to pave the way for success. For example, many candidate HIV vaccines have the potential to cause vaccine-induced seropositivity (or seroreactivity), which can interfere with the correct interpretation of routine HIV diagnostic testing. Vaccine developers will now need to work together with diagnostic developers to generate new methods that avoid the misclassification of HIV diagnoses because of HIV vaccination [134]. In addition, the COVID-19 pandemic has shown to us that vaccine misinformation can greatly reduce the uptake of even a highly safe and effective product. A recent scoping review found that incorrect beliefs surrounding HIV vaccination were already widely prevalent in multiple different sampled areas, and misinformation was most likely to be identified among high-incidence populations, compared to low-incidence populations [135]. Another study also found that reported facilitators for a hypothetical HIV vaccine included improved knowledge and understanding, particularly of their own risk of acquiring HIV [15]. Efforts to alleviate HIV vaccine hesitancy and build trust in impacted communities should be emphasized, in parallel with the intensive discovery medicine program discussed in this review.

## 7. Conclusions

It has been over forty years since the surge in cases of *Pneumocystis* pneumonia and Kaposi sarcoma across the United States prompted the federal government’s first investigation into the HIV pandemic [136]. At that time, the effort to diagnose, treat, and prevent HIV infection was one of the world’s greatest public health undertakings. The approval of the first “HTLV-3” antibody test in 1985, the advent of AZT in 1987, and the introduction of highly active antiretroviral therapy in 1996 were just the first steps in a long march that has brought us closer to the dream of ending the HIV epidemic [137]. While ART has revolutionized the treatment and prevention of HIV, it is a solution that requires lifelong access and adherence to medication. The key to HIV eradication is a safe and effective vaccine that prevents new infections, reduces spread, and addresses disparities among marginalized and vulnerable populations. The ability of HIV to evade the immune system has posed substantial challenges to vaccine development. However, recent trials have uncovered key insights into viral and immune biology, bringing about novel strategies for immunogen design. The most promising of the next wave of clinical trials in HIV vaccine development are those that integrate the various disciplines of thought in immunogen design and platform, utilizing the overlapping strategies of lineage maturation, germline targeting, and structurally informed epitope design, while also further exploring mRNA platforms and adjuvant selection. Furthermore, the use of viral vectors to stimulate robust T cell responses in ways that emulate natural infection has provided a promising avenue for inducing sustained immunity to or control of HIV. Continued research and integrative efforts are essential to refine these approaches and translate them into an effective HIV vaccine.

## Figures and Tables

**Table 1 vaccines-12-01043-t001:** Recent phase 3 efficacy trials in HIV vaccine or antibody-mediated prevention.

Trial ID	ClinicalTrials.gov ID	Immunogen
HVTN 702 (Uhambo)	NCT02968849	ALVAC-HIV plus bivalent subtype C gp120–MF59
HVTN 703/HPTN 084 (AMP)	NCT02568215	VRC01 neutralizing antibody
HVTN 704/HPTN 085 (AMP)	NCT02716675	VRC01 neutralizing antibody
HVTN 705 (Imbokodo)	NCT03060629	Ad26.Mos4.HIV
HVTN 706 (Mosaico)	NCT03964415	Ad26.Mos4.HIV

**Table 2 vaccines-12-01043-t002:** HIV vaccine development: new B and T cell approaches in preventive vaccine clinical trials.

Approach	Strategy	Example Recent/Ongoing Clinical Trials
Overlapping B cell approaches	Goal: to elicit broadly neutralizing antibody (bnAb) responses against HIV-1	
Mutation-guided immunogen design	A priming immunogen is identified that has affinity for the unmutated common ancestor of a known bnAb. Repeated immunization with additional soluble HIV-1 Env constructs leads to affinity maturation.	HVTN 115 [NCT03220724] HVTN 137 [NCT04177355] HVTN 300 [NCT04915768] HVTN 302 [NCT05217641] HVTN 307 [NCT05903339]
Structure-based immunogen design	A priming immunogen is identified that binds diverse precursors within a bnAb class that spans many lineages.	IAVI G001 [NCT03547245] IAVI G002 [NCT05001373] HVTN 807 [NCT06006546]
Germline/lineage-agnostic immunofocusing approach	Immunogens are identified that stimulate broadly neutralizing epitope-targeted antibodies; often have affinity for unmutated common ancestors.	HVTN 133 [NCT03934541]
Novel T cell approach	Goal: to elicit MHC-E-restricted CD8+T cells to protect against HIV-1 by arresting and clearing a new infection	
Arrest and clear infection	Live, attenuated recombinant human cytomegalovirus (HCMV)-vectored HIV-1 vaccine.	VIR-1111 [NCT04725877] HVTN 142 [NCT05854381]

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
