# Peer review of "HIV Vaccine Development at a Crossroads: New B and T Cell Approaches"

_vaccines, 2024, doi:10.3390/vaccines12091043_

Round 1

Reviewer 1 Report

Comments and Suggestions for Authors

This is an interesting review article on HIV vaccine development. The writing is mostly good and clear. The introduction that makes the case for why an HIV vaccine is still needed is exceptionally compelling, and the conclusion likewise is very clear and well written. However, parts of the meat of the article are sometimes confusing and missing some key concepts and references. The article starts appropriately with a discussion of neutralizing antibodies and the history of diversion from that (which is the usual goal of vaccine development) by the RV144 trial results that pinned many investigators’ hopes on never clearly defined nor agreed upon non-neutralizing antibodies; this discussion is reasonable and appropriately avoids the sometimes heated arguments of the proponents and detractors of the non-neutralizing antibody approach. The article then goes on to discuss the goal of inducing broadly neutralizing antibodies in more detail. This discussion is a bit unclear and is missing a few key articles. First of all, long before the AMP trial, the ability of neutralizing antibodies to protect against SHIV challenge in nonhuman primates was demonstrated by investigators at the Walter Reed Army Institute of Research; this article [Mascola J.R., Stiegler G., VanCott T.C., et al. Protection of macaques against vaginal transmission of a pathogenic HIV-1/SIV chimeric virus by passive infusion of neutralizing antibodies. Nat Med. 2000;6(2):207-10] should be cited as it really established the basis for developing a neutralizing antibody-based vaccine for HIV. Another article that should be cited is the paper by Dimitrov et al [Xiao X., Chen W., Feng Y. and Dimitrov D.S. Maturation pathways of cross-reactive HIV-1 neutralizing antibodies. Viruses 2009 1:1802-817] which was the landmark, first description of the fact that broadly neutralizing antibodies against HIV-1 have all been subject to much more extensive somatic hypermutation than neutralizing antibodies against other pathogens. This observation was key to understanding that, unlike all other diseases where there were neutralizing antibodies in peoples’ antibodyome to be induced with a properly constructed immunogen, the challenge of an HIV vaccine is to evolve the broadly neutralizing antibodies required for protection against an extremely diverse population of envelope variants from narrowly specific and/or poorly binding antibody genes in naïve B cells. And in discussion of broadly neutralizing antibodies confusion is introduced by a bit of a conflation of clade with serotype; there are no serotypes in HIV as there are with many other pathogens because despite the extreme diversity of sequence in the HIV Envelope protein, bNAbs target the same 5 or 6 epitopes that are present in all clades with sequence diversity not changing the epitope but rather subtly changing access to the epitopes (which should be thought of as at the bottom of wells formed by flexible glycan chains which the antibody Fab must navigate to get to the epitope). The article’s discussion of the different strategies to induce bNAbs is a bit muddled but it is hard to completely fault the authors because the literature is a bit muddled partly because investigators have developed different terms to distinguish themselves from others rather than because there are fundamental differences between what they are actually doing. The lineage-based design advocates may start with a transmitted founder virus envelope from the lineage of bNAbs they seek to induce but then they open it up structurally to bind better to the germline genes they want to target in ways similar to what the germline targeting camp does; then they usually use structural modifications to design their boosts rather than envelopes from elsewhere in the lineage; and both camps believe that a basic problem is the limited number of germline antibody genes that can evolve into bNAbs, so whether that number is 3 or 17 it is still many fewer than for neutralizing antibodies against other pathogens so they are all in a sense “germline-targeting.” A better description of the different strategies can be found in the article from the HVTN investigators on their Discovery Medicine phase 1 clinical trials [Martin T.M., Robinson S.T. and Huang Y. Discovery medicine– the HVTN's iterative approach to developing an HIV-1 broadly neutralizing vaccine. Curr Opin HIV AIDS. 2023;18(6):290–299]. Also, the “epitope-focused” strategy is well described but then the authors fail to comment on the fact that the MPER immunogen in the very promising HVTN133 trial is simply a peptide stuck in a liposome and not really “rooted in technical understandings of Env secondary and tertiary structure” (which is really much more descriptive of the germline targeted eOD-GTs); it might be better just to describe the epitope-focused strategy as immunizing with the targeted epitope separated from the rest of the envelope. The last paragraph in this section is a good explanation of the hoped-for contribution of mRNA vaccine technology to HIV vaccine development. The last section of the review is entitled “T Cell Approaches to HIV Vaccine Development” which is a bit misleading as it only discusses one T cell approach (a CMV-vectored vaccine) not “approaches” and it doesn’t even mention the most remarkable aspect of this vaccine which is that it is based on CD8+ T cells recognizing antigen presented by MHC-E rather than the usual class I MHC molecules. This section is both skimpy in example and discussion of underlying concepts (any serious discussion of T cell-based approaches to an HIV vaccine should discuss or at least mention that a primary obstacle to such approaches is the extreme epitopic diversity of T cell targets in the HIV genome, many of which are extremely easy for the virus to escape from a T cell response by low fitness cost mutations; and the problem is compounded by the population diversity of MHC class I molecules which means that each individual presents a different subset of the great diversity of T cell epitopes). There have been many different approaches to dealing with the epitopic diversity in HIV from sequence based (mosaic and conserved) to epitope selection (functional, protease cleavage site, networked), but the CMV vaccine of Picker ignores the problem completely and has just shown protection in a homologous challenge model where the only sequences present in the challenge virus were those present in the vaccine. So not only is the T cell-based vaccines section very skimpy but the only vaccine approach it discusses doesn’t even attempt to deal with a major challenge in HIV for T cell-based vaccines. There is no mention of the failed STEP efficacy trial which stifled T cell-based HIV vaccine approaches for a decade. Nor is there any mention of the several interesting attempts at T cell induction in people living with HIV by Brander, Hanke and Korber, and Goonetileke, or the pioneering work in this field by Andrew McMichael and Bruce Walker (the director of the Ragon Institute where the senior author of this review article works) or the innovative nonhuman primate studies by Masopust. The very last section in this article (HIV Vaccine Challenges Beyond Immunogen Design) by addressing VISP and vaccine hesitancy covers some important topics not usually discussed and therefore is appreciated by this reviewer. Lastly, I would like to point out one unfortunate apparent typo in line 151 where the word “virus” is substituted for the word “vaccine” resulting in a very silly statement. All in all though this is a very good start at nice review article which can easily be made acceptable for publication with the addition of some more detail in some sections, correction of some minor confusions, and addition of some important references.

Author Response

Please see attached "Reviewer Response Letter" and refer to our responses to "REVIEWER 1". 

Reviewer 2 Report

Comments and Suggestions for Authors

Manuscript is well written and covers most of the recent developments in the field.

It would have been good to also include the details on stem cell transplantation where few cases have been reported with complete virus clearance in the introduction section.

Author Response

Please see the attached "Reviewer Response Letter" and refer to responses to "REVIEWER 2". 

Reviewer 3 Report

Comments and Suggestions for Authors

The authors have submitted the review manuscript titled "HIV Vaccine Development at a Crossroads: New B and T Cell Approaches". In this, the authors describe the B and T-cell based approaches including epitope-focused, germline targeting etc. in the development of vaccines against HIV. Overall, the manuscript is very well-written and thoughroughlly analyzes the available literature and addresses the gaps in research. 

I only have 1 minor comment. I think that the authors should put the phase 3 efficacy trials in the tabular format.

Author Response

Please see the attached "Reviewer Response Letter" and refer to responses to "REVIEWER 3."

Round 2

Reviewer 1 Report

Comments and Suggestions for Authors

The authors have responded appropriately to all of my earlier criticisms and I find the revised version of the manuscript completely acceptable. Good job.